# Deep learning to predict long-term mortality in patients requiring 7 days of mechanical ventilation

**Naomi George** [1,2]*, **Edward Moseley**[3], **Rene Eber** [4,5], **Jennifer Siu**[2,6], **Mathew Samuel**[2], **Jonathan Yam**[2], **Kexin Huang**[2], **Leo Anthony Celi**[2,4,7], **Charlotta Lindvall**[3,8]

**1** Department of Emergency Medicine, Division of Critical Care, University of New Mexico Health Science Center, Albuquerque, New Mexico, United States of America, **2** Harvard T.H. Chan School of Public Health, Boston, Massachusetts, United States of America, **3** Department of Psychosocial Oncology and Palliative Care, Dana-Farber Cancer Institute, Boston, Massachusetts, United States of America, **4** Massachusetts Institute of Technology, Cambridge, Massachusetts, United States of America, **5** Université de Montpellier, Montpellier, France, **6** Department of Otolaryngology, Division of Head & Neck Surgery, University of Toronto, Toronto, Canada, **7** Department of Medicine, Beth Israel Deaconess Medical Center, Boston, Massachusetts, United States of America, **8** Department of Medicine, Brigham and Women's Hospital, Boston, Massachusetts, United States of America

* nageorge@salud.unm.edu

**Data Availability Statement:** The data underlying the results presented in the study are available from MIMIC: https://github.com/MIT-LCP/mimic-code.

## Abstract

### Background

Among patients with acute respiratory failure requiring prolonged mechanical ventilation, tracheostomies are typically placed after approximately 7 to 10 days. Yet half of patients admitted to the intensive care unit receiving tracheostomy will die within a year, often within three months. Existing mortality prediction models for prolonged mechanical ventilation, such as the ProVent Score, have poor sensitivity and are not applied until after 14 days of mechanical ventilation. We developed a model to predict 3-month mortality in patients requiring more than 7 days of mechanical ventilation using deep learning techniques and compared this to existing mortality models.

### Methods

Retrospective cohort study. Setting: The Medical Information Mart for Intensive Care III Database. Patients: All adults requiring $\geq$ 7 days of mechanical ventilation. Measurements: A neural network model for 3-month mortality was created using process-of-care variables, including demographic, physiologic and clinical data. The area under the receiver operator curve (AUROC) was compared to the ProVent model at predicting 3 and 12-month mortality. Shapley values were used to identify the variables with the greatest contributions to the model.

### Results

There were 4,334 encounters divided into a development cohort (n = 3467) and a testing cohort (n = 867). The final deep learning model included 250 variables and had an AUROC of 0.74 for predicting 3-month mortality at day 7 of mechanical ventilation versus 0.59 for the

**Funding:** The authors received no specific funding for this work.

**Competing interests:** The authors have declared that no competing interests exist.

ProVent model. Older age and elevated Simplified Acute Physiology Score II (SAPS II) Score on intensive care unit admission had the largest contribution to predicting mortality.

## Discussion

We developed a deep learning prediction model for 3-month mortality among patients requiring ≥ 7 days of mechanical ventilation using a neural network approach utilizing readily available clinical variables. The model outperforms the ProVent model for predicting mortality among patients requiring ≥ 7 days of mechanical ventilation. This model requires external validation.

## Introduction

Nearly 70% of older adults report that they prioritize quality of life over longevity [1]. Many would prefer death over prolonged survival dependent on mechanical ventilation (MV) [2–5]. Yet, increasingly older adults with acute respiratory failure are treated with MV [6, 7]. By 2020, more than half of the estimated 600,000 critically ill patients treated with MV will be older adults (age ≥ 65 years), and approximately 20% will subsequently undergo tracheostomy [8–13], making tracheostomy one of the most common elective procedure in the intensive care unit (ICU) [14]. Outcomes among older adults who receive tracheostomy are poor; by 12-months 60–70% will have died, and fewer than 10% will have achieved functional independence [10, 12, 13, 15–18]. Identifying the subset of patients who are likely to benefit from tracheostomy, and those who will not, continues to pose a substantial challenge to clinicians [19].

For patients with acute respiratory failure, MV is initially delivered via an endotracheal tube which is passed through the oral cavity. If the respiratory function does not improve after a period of 7–21 days, then the endotracheal tube is often replaced [surgically or percutaneously] with a tracheostomy tube [20–24]. Among patients who survive their acute illness and regain good function, tracheostomy can increase comfort, decrease delirium during MV, and facilitate faster recovery from MV [20–25]. Paradoxically, among the many patients who ultimately do not survive, tracheostomy may serve only to prolong the dying process [15–17, 26]. The toll falls particularly hard on older adults, many of whom will suffer from high rates of distressing symptoms associated with chronic critical illness for weeks or months after tracheostomy but prior to death. Others risk becoming trapped in a state of chronic critical illness, enduring a prolonged but often dismal survival [15–17, 26]. Moreover, healthcare costs and resource utilization associated with the care of older adults who receive tracheostomy are staggering, and do not meet standard thresholds of acceptability for cost effectiveness [27]. Unfortunately, patient's surrogate decision-makers frequently receive little information regarding the probability of long-term survival and good functional outcome, and often have unrealistic expectations regarding survival [12, 28]. This is due in part clinicians lack of awareness of and comfort with expected prognosis [29–31].

Existing general ICU mortality prediction models, such as the Acute Physiology and Chronic Health Evaluation (APACHE) score, were developed to predict in-hospital mortality and perform poorly in predicting long term survival of patients requiring prolonged mechanical ventilation [32, 33]. Clinical tools such as the ProVent Score can be used to predict 1-year mortality after 14 or 21 days of MV. Such tools were developed with the intention of informing prognosis in decision-making conversations around tracheostomy [34–38]. However, multiple randomized trials have demonstrated that 'early' tracheostomy placement (< 10 days after

initiation of MV) results in decreased need for sedatives, fewer days of MV, decreased ICU stay, and may be associated with decreased long-term mortality as compared to 'late' (>10 days) tracheostomy placement [22, 39]. Thus, more than 50% of tracheostomies are now placed early (<10–14 days) [39], at which time the validity of the ProVent Score is unknown. Among patients with poor prognosis, enduring 14–21 days of ICU care prior to prognostication may be unnecessarily burdensome.

To address this gap, we sought to develop a mortality prediction model to enhance decision making around tracheostomy. Our objective was to develop and validate deep learning model to predict 3- and 12-month mortality among patients requiring more than 7 days of mechanical ventilation for acute respiratory failure.

## Methods

### Design, setting, and population

Data were obtained from the Medical Information Mart for Intensive Care III (MIMIC-III) database. MIMIC-III contains records of 61,051 ICU admissions at Beth Israel Deaconess Medical Center in Boston, Massachusetts from June 2001, through October, 2012 [40]. Inclusion criteria included patients over the age of 18 years admitted to the neurological, trauma, surgical, cardiac, or medical ICU and who were treated with MV for $\geq$ 7 days. Patients were excluded if they had a primary hospital diagnosis of head and neck cancer requiring surgical intervention in the neck, burns comprising >30% body area, burns involving the head and neck, or acute neuromuscular disorders (e.g. Guillain Barre) (S1 Table). The data in MIMIC-III has been previously de-identified, and the institutional review boards of the Massachusetts Institute of Technology (No. 0403000206) and Beth Israel Deaconess Medical Center (2001-P-001699/14) both approved the use of the database for research. The Guidelines for Developing and Reporting Machine Learning Predictive Models in Biomedical Research were followed throughout this project [41].

### Feature selection and data processing

Processes-of-care variables selected for inclusion in the model were reviewed by meidcal clinicians (NG, CL, JS) and from existing ICU mortality prediction literature and included all the variables from the ProVent Model (Fig 1). Demographic variables were carefully selected to avoid overt reinforcement of health disparities. Thus race, ethnicity, insurance status, zip code were all excluded from the model.

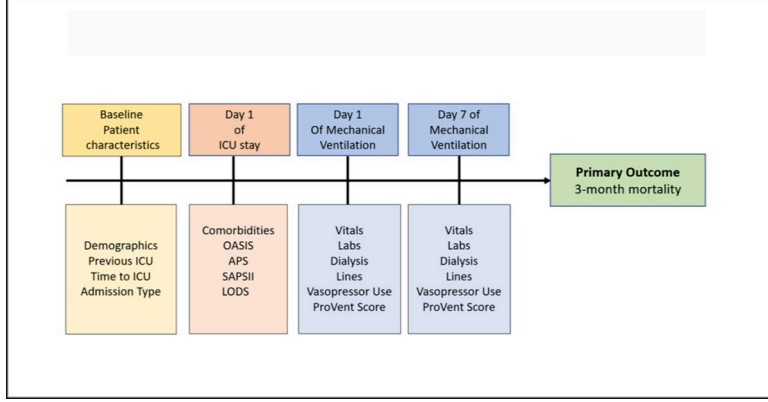

**Fig 1. Variable selection for machine learning neural network prediction model.**

The extracted data contained both static information, (e.g. age, sex, duration of hospitalization prior to ICU admission, reason for admission, site of admission, and International Classification of Diseases, 9th Revision codes), as well as temporal and dynamic data (e.g. time-stamped laboratory values, vital signs, medication administrations). Feature engineering was performed on continuous variables (e.g. maximum value, minimum value, and mean value). Only variables obtained from the first 7 days of MV were included. Continuous variables were standardized to a mean of zero and scaled to unit variance. Additionally, five severity of illness scores were included as predictors: the Oxford Acute Severity of Illness Score (OASIS), the Simplified Acute Physiology Score (SAPS), the Simplified Acute Physiology Score Version II (SAPSII), the Acute Physiology Score III (APSIII), and the Logistic Organ Dysfunction System (LODS) [42–47]. The Elixhauser van Walraven comorbidity score and its 30 component items were also included [48]. Mortality data was obtained from the Social Security Death Index.

## Model building and validation

The dataset was randomly divided into a training (80%) and a testing set (20%). A multi-layer feedforward neural network was created. Neural networks approximate the best separating function for labeled input data and can learn any arbitrary, complex functions. They do not use weight-sharing across layers; information flows in one-direction, from the input layers, through intermediate hidden layers to the output layer. A sigmoid activation function was used in the output layer, with which the output range was interpreted as a prediction probability of the primary outcome, 3-month mortality (**S1 Fig**). For better generalization capabilities, we utilized dropout and L1-L2 regularization in each hidden layer. These techniques prevent complex co-adaptations on training data, preventing overfitting. For missing or outlier values of continuous data, normal values were imputed. Individual outliers were reviewed and discarded if deemed erroneous.

## Statistical analysis and outcomes

The primary outcome was 3-month mortality. The final model chosen was based on the highest calibration determined by area under the receiver operator curve (AUROC) in the testing set. The secondary outcome was 12-month mortality. Sensitivity, specificity, AUROC, accuracy and F1 score are also reported. Shapley values, which reveal the marginal contribution of the individual variables across permutations, were reported to facilitate understanding of the model [49]. The neural network models' predictive ability was compared to the ProVent model. The ProVent model was developed to predict mortality among patients requiring MV for 14 days or more, and has been validated in several studies since its publication [34–37, 50]. We calculated the performance of the ProVent score using the component variables as reported in the original manuscript. In addition, in order to give the logistic regression model the 'best chance' to compete with the deep learning model we also built 'extended' logistic regression models using the ProVent variables as well as each of the other variables available to the deep learning model using 10 fold cross-validation (*extended LR*) and least absolute shrinkage and selection operator (*extended LR LASSO*) techniques. The AUROC for was calculated for ProVent logistic regression based on values at 7 days of MV. Statistical analysis and model building was performed using Python v3.6 [51].

## Results

A total of 61,532 unique ICU stays were identified within the MIMIC database. Of those, there were 4,334 unique ICU stays representing 3,982 unique ICU patients receiving ≥ 7 days of MV and meeting the inclusion and exclusion criteria. Patient selection is outline in **Fig 2**.

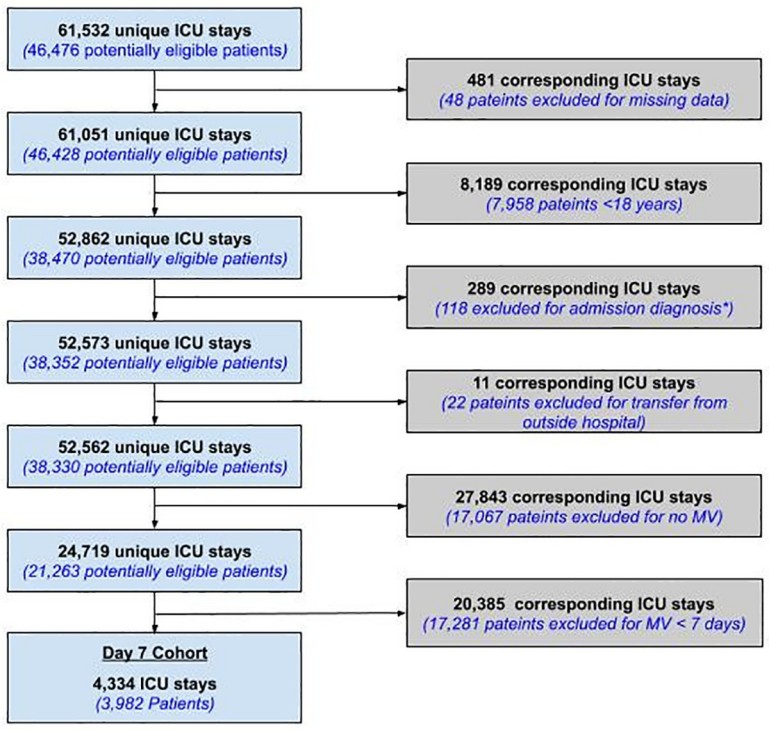

**Fig 2. Flow diagram.**

Eighty percent (n = 3,467) of this study sample were used as the initial training data set and the remaining 20% (n = 867) were used to test the prediction models.

Of the 3,982 patients that met inclusion criteria 59.6% were alive at 3 months. Baseline patient characteristics are summarized in **Table 1**. The median age was 65.9 years (interquartile range (IQR) 52.7, 76.8). The majority of patients (70.9%) were white, admitted to the ICU from the emergency room 49.4%. Median ICU length of stay was 15.7 days (IQR of 10.8, 22.5). The development and testing cohorts did not significantly differ in terms of illness severity or baseline comorbidities with the exception of baseline chronic obstructive pulmonary disease and metastatic cancer. Laboratory values, vital signs, and clinical interventions were not significantly different between development and testing sets (**S2 Table**).

The final deep learning model included 250 variables. The performance of the neural network and the ProVent Model for prediction of 3- and 12-month mortality at day is shown in **Table 2**. The performance of the model for predicting 3-month and 12-month mortality status is shown in **Fig 3**. The day 7 neural network model had an AUROC of 0.74 for 3-month mortality and 0.76 for 12-month mortality, versus 0.59 and 0.63 for the ProVent model, respectively. Of note, in our cohort, the ProVent Model performed worse when measured at 7 days (AUC 0.59) than in previously published studies. The positive and negative predictive value for the neural network at 3 months was 0.64 and 0.72 respectively, and 0.67 and 0.71 at 12 months. The calibration curve for the testing and training set for both 3-month and 12-month mortality are shown in **S2 Fig**, and the area under the precision recall curves are shown in **S3 Fig**. In addition, comparison of the neural network model with the extended LR and extended LR LASSO models are available in **S3 Table**.

Analysis of relative variable importance to the model prediction using shapley values showed that increased use of renal replacement therapy on day 7 of MV, elevated sodium levels

**Table 1. Patient characteristics.**

| | Training n = 3,467 | Testing n = 867 | p-value |
|---|---|---|---|
| **Sex** | | | |
| Female, n (%) | 1494 (43.1) | 361 (41.6) | 0.462 |
| **Age (years)** | | | |
| Median (IQR) | 65.8 (52.9, 76.8) | 65.9 (52.2, 76.9) | 0.694 |
| **Race/Ethnicity n (%)** | | | |
| Asian | 77 (2.2) | 14 (1.6) | 0.661 |
| Non-Hispanic Black | 277 (8.0) | 73 (8.4) | |
| Hispanic | 91 (2.6) | 19 (2.2) | |
| Unknown/Others | 562 (16.2) | 150 (17.3) | |
| Non-Hispanic White | 2460 (71.0) | 611 (70.5) | |
| **Source Location of ICU Admission, n (%)** | | | |
| Emergency Department | 428 (49.4) | 1724 (49.7) | 0.397 |
| Office Referral | 218 (25.1) | 928 (26.8) | |
| Transfer from Hospital or SNF | 221 (25.5) | 815 (23.5) | |
| **First Care Unit, n (%)** | | | |
| Coronary Care Unit | 355 (10.2) | 94 (10.8) | 0.374 |
| Cardiac Surgery Recovery Unit | 436 (12.6) | 108 (12.5) | |
| Medical Intensive Care Unit | 1382 (39.9) | 372 (42.9) | |
| Surgical Intensive Care Unit | 706 (20.4) | 162 (18.7) | |
| Trauma/Surgical Intensive Care Unit | 588 (17.0) | 131 (15.1) | |
| **Illness Severity on ICU Admission (median, IQR)** | | | |
| SOFA Score | 6.0 (4.0, 9.0) | 6.0 (4.0, 9.0) | 0.43 |
| LODS | 6.0 (4.0, 8.0) | 6.0 (3.0, 8.0) | 0.736 |
| OASIS | 38.0 (32.0, 44.0) | 38.0 (33.0, 44.0) | 0.286 |
| SAPS II | 42.0 (33.0, 52.5) | 43.0 (34,0, 54.0) | 0.227 |
| **Elixhauser Comorbidities, n (%)** | | | |
| Elixhauser Score, (median, IQR) | 4.0 (2.0, 5.0) | 4.0 (2.0, 5.0) | 0.41 |
| Congestive Heart Failure | 1227 (35.4) | 325 (37.5) | 0.267 |
| Chronic Pulmonary Disease | 881 (25.4) | 239 (27.6) | 0.21 |
| Liver disease | 562 (16.2) | 142 (16.4) | 0.945 |
| Renal Failure | 520 (15.0) | 117 (13.5) | 0.287 |
| Metastatic Cancer | 131 (3.8) | 33 (3.8) | 1.000 |
| **Mortality, n%** | | | |
| Time to death, days (median, IQR) | 55.1 (18.2, 314.0) | 52.2 (18.2, 209.9) | 0.309 |
| 3-month Mortality | 1390 (40.1) | 360 (41.5) | 0.466 |
| 12-month Mortality | 1714 (49.4) | 428 (49.4) | 1.000 |
| **Length of Stay (LOS), median (IQR)** | | | |
| Hospital LOS | 22.57 (15.7, 32.3) | 22.32 (15.7, 31.6) | 0.713 |
| ICU Length of Stay | 15.69 (11.6, 23.1) | 15.78 (11.5, 23.8) | 0.863 |
| **Hospital Disposition, n (%)** | | | |
| Died | 979 (28.2) | 250 (28.8) | 0.651 |
| Home / Home health care | 396 (11.4) | 111 (12.8) | |
| Hospice | 26 (0.7) | 9 (1.0) | |
| Long Term Acute Care | 468 (13.5) | 106 (12.2) | |
| Short Term Hospital/other | 40 (1.2) | 12 (1.4) | |
| Subacute Nursing Facility (SNF) | 1558 (44.9) | 379 (43.7) | |

**Table 2. Deep model performance versus ProVent.**

| Model | Three-Month Mortality | 12-Month Mortality |
|---|---|---|
| **Deep Learning at 7 Days** | | |
| Area under ROC | 0.74 | 0.76 |
| Sensitivity | 0.58 | 0.72 |
| Specificity | 0.76 | 0.65 |
| Accuracy | 0.69 | 0.69 |
| F1 Score | 0.61 | 0.69 |
| **ProVent at 7 Days** | | |
| Area under ROC | 0.59 | 0.63 |
| Sensitivity | 0.41 | 0.73 |
| Specificity | 0.78 | 0.53 |
| Accuracy | 0.63 | 0.63 |
| F1 Score | 0.47 | 0.66 |

on day 1 of MV, increased age, and increased heart rate on day 7 of MV had the largest impact on the model's prediction of 3-month. Similarly, increased age, increased heart rate on day 7 of MV, use of renal replacement therapy on day 7 of MV, and low diastolic blood pressure on day 7 had the greatest impact on the model's prediction of 12-month mortality (**Fig 4**).

## Discussion

We demonstrate that a feedforward neural network model, based on clinical variables readily available during routine processes-of-care in the ICU, can accurately predict 3- and 12-month mortality among ICU patients requiring mechanical ventilation for $\geq 7$ days. Our model had superior performance at 7 days compared to one of the most commonly used mortality prediction models for mechanically ventilated patients, the ProVent Score, and can be applied earlier in a patients' ICU course.

Predicting which ICU patients are at high risk for a poor outcome is essential in order to make more informed decisions regarding medical interventions at the end of life, and minimize suffering for patients who are on a dying trajectory [15, 52, 53]. However, mortality risk prediction models among ICU patients are infrequently used to inform clinical decisions [12, 28–31]. This is attributable to several factors. First, the most common ICU mortality prediction models, such as the APACHE, SAPS, LODS, and MPM scores, are derived from data

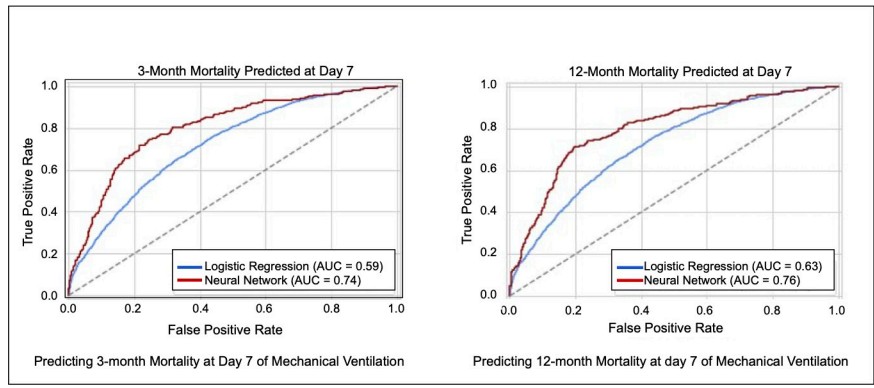

**Fig 3. Predicting 3- and 12-month mortality at day 7 of mechanical ventilation.**

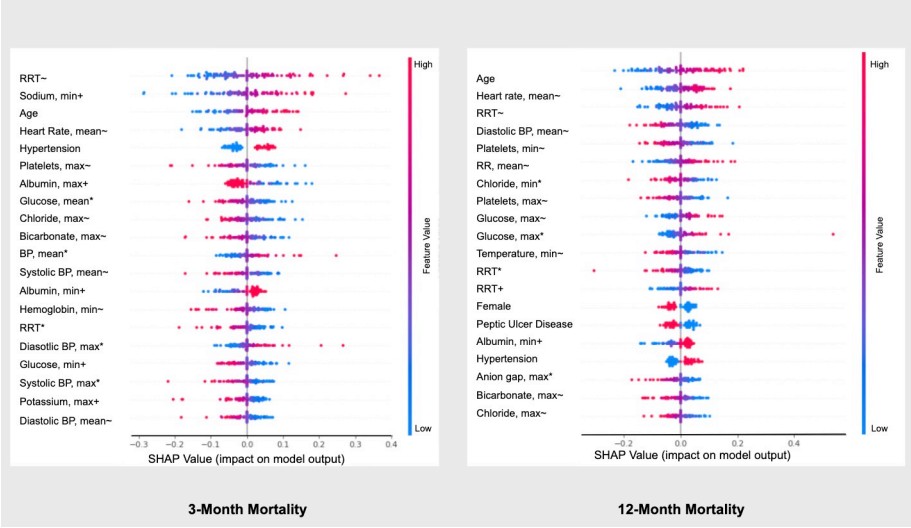

**Fig 4. SHAP values for 3 and 12-month mortality.** *value on ICU day 1 of ICU; + value on day 1 mechanical ventilation; ~ value on day 7 of mechanical ventilation; RRT: Renal Replacement Therapy; min: minimum value; max: maximum value; BP: blood pressure; RR: respiratory rate.

obtained in the first 24 hours of the ICU stay. Yet, many patients and families prefer to attempt a trial of therapy (typically several days in duration), prior to goals of care decision-making, after which predictions gleaned during the first 24 hours may no longer accurately reflect the prognosis. Second, most mortality prediction models, including the SOFA score and APACHE score, are calibrated to predict in-hospital mortality [32, 33, 43]. However, post-discharge prognosis can also contribute to more meaningful discussions with patients.

Among patients surviving ≥ 7 days of mechanical ventilation, one of the most critical decisions clinicians, patients, and families will face is whether or not to undergo tracheostomy. Among patients on mechanical ventilation with a good long-term prognosis, transition to tracheostomy can improve comfort, facilitate early mobility, and enhance recovery [54]. However, for older adult ICU survivors, those requiring ongoing treatment with mechanical ventilation often face a dismal quality of life, dying within a year of tracheostomy placement [55].

For most ICU patients requiring MV the decision of whether to undergo tracheostomy falls to a surrogate decision maker(s), who in turn relies on the ICU clinicians to share information about prognosis and guide expectations. Surrogate decision-makers often seek prognostic disclosures from clinicians [30]. However, studies have shown significant problems with communication of ICU prognosis; information may be subject to bias and is often poorly communicated by clinicians, misunderstood by families, or never disclosed at all [12, 28]. Clinicians themselves have only modestly accurate prognostic ability in terms of mortality–often overestimating the probability of a survival [31, 56]. If surrogate decision-makers of patients with poor prognosis were made aware of risk for poor outcome, it is likely many would not choose tracheostomy and ongoing MV [2, 57].

It is imperative that clinicians have prognostic information about the probability of survival after tracheostomy prior to placement of tracheostomy in order to facilitate decision-making. Among long-term survivors, delay in tracheostomy decision past 10 days may threaten the patient's ability to maximally benefit from tracheostomy, and increases the duration of burdensome symptoms associated with endotracheal tubes. Among decedents, the delay in tracheostomy decision may mean a missed opportunity for earlier transition to hospice care or

natural death with less suffering than results from continued ICU care with mechanical ventilation.

Severity scores like ProVent are based on logistic regression models that assume linear and additive relationship of predictors and outcomes. However, these assumptions may not be valid in the context of critical illness with very complex underlying processes and complicated interactions between patient factors, disease features and treatments administered. Indeed, in our study the ProVent score performed poorly as compared to the machine learning model. This is likely due in part to the fact that the ProVent model was not developed for application at day 7, and also due to the strengths of machine learning. Machine learning methods can offer a more flexible statistical approach and have been shown to outperform conventional logistic regression statistical models for several medical conditions [58, 59]. Interestingly, in our study the LASSO model, which is a hybrid between regression and machine learning techniques, significantly outperformed the ProVent model giving results similar to the neural network.

In terms of accuracy, our model compares similarly to other machine learning mortality prediction models developed for critically ill patients. Dybowski et. al. used a cohort of 258 ICU patients to create an artificial neural network enhanced by generic algorithms achieving an AUC 0.86, however their study was limited to a small subset of patients who either had a systemic inflammatory response syndrome or hemodynamic shock [60]. Pirracchio et al developed a "Super Learner" model using ensemble machine learning technique with multiple learning algorithms to predict in-hospital mortality [58]. Kim et. al compared decision trees, artificial neural networks, and support vector machine models predicting mortality in pediatric ICU patients, achieving an AUC 0.87–0.89, outperforming traditionally used regression models [61].

To the best of our knowledge our model is the first to predict long-term mortality among mechanically ventilated patients $\geq$ 7 days of MV. Our approach is strengthened in the use multiple time points within a patient's clinical trajectory (baseline variables collected at ICU admission, then day 1 of MV, and day 7 MV) which allows for flexibility and time-sequence analysis in the event of patient status changes during their clinical course.

There are limitations to this study, and the results should only be interpreted in the context of its study design. All database analyses are susceptible to coding misclassification and bias from error. Further, the data comes from a single institution, whose organization and clinical practice patterns may differ from other institutions, limiting the generalizability of our model. In addition, the data in MIMIC only extends to 2012. Our data was compared to the ProVent prediction model which was optimized for day 14 and 21 of mechanical ventilation. Because we believe day 7 on mechanical ventilation to be more clinically relevant, this comparison may be limited. Nonetheless this study represents an important methodology to build a clinical tool that can provide more detailed insight into the prognosis of patients who may require tracheostomy.

## Conclusion

Here we demonstrate the ability of using a neural network to predict 3- month mortality in patients on mechanical ventilation for more than 7 days. Deep learning prediction models are becoming increasingly important in our data-driven clinical decision-making, especially in the absence of randomized controlled trials. Further optimization of these by external validation, prospective validation using external cohorts, and subgroup analysis within specific populations is integral prior to widespread implementation. Ultimately, integration of deep learning prediction models like ours into electronic health records will provide valuable information to enable providers and patients to more informed decisions.

## Supporting information

**S1 Fig. Schematic diagram of forward feed neural network model.**
(TIF)

**S2 Fig. 3 and 12-month calibration curve.**
(TIF)

**S3 Fig. 3- and 12-month area under the precision recall curve.**
(TIF)

**S1 Table. Exclusion criteria ICD-9 code.**
(TIF)

**S2 Table. Vital signs and laboratory values.**
(TIF)

**S3 Table. Deep model performance versus ProVent.**
(TIF)

## Acknowledgments

Collaborative Data Science for Medicine Course, in the Laboratory for Computational Physiology at the Massachusetts Institute of Technology.

## Author Contributions

**Conceptualization:** Naomi George, Jennifer Siu, Mathew Samuel, Jonathan Yam, Leo Anthony Celi, Charlotta Lindvall.

**Data curation:** Naomi George, Edward Moseley, Rene Eber, Jennifer Siu, Mathew Samuel, Jonathan Yam.

**Formal analysis:** Naomi George, Edward Moseley, Rene Eber, Kexin Huang, Charlotta Lindvall.

**Investigation:** Naomi George.

**Methodology:** Naomi George, Edward Moseley, Charlotta Lindvall.

**Project administration:** Naomi George, Edward Moseley, Charlotta Lindvall.

**Resources:** Leo Anthony Celi.

**Supervision:** Naomi George, Leo Anthony Celi, Charlotta Lindvall.

**Writing – original draft:** Naomi George, Edward Moseley, Rene Eber, Jennifer Siu, Mathew Samuel, Jonathan Yam, Kexin Huang, Leo Anthony Celi, Charlotta Lindvall.

**Writing – review & editing:** Naomi George, Edward Moseley, Jennifer Siu, Mathew Samuel, Jonathan Yam, Kexin Huang, Leo Anthony Celi, Charlotta Lindvall.

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
