## [Decision Letter · Decision Letter 0]

16 Dec 2020

PONE-D-20-32346

Deep learning to predict long-term mortality in patients requiring 7 days of mechanical ventilation

PLOS ONE

Dear Dr. George,

Thank you for submitting your manuscript to PLOS ONE. After careful consideration, we feel that it has merit but does not fully meet PLOS ONE’s publication criteria as it currently stands. Therefore, we invite you to submit a revised version of the manuscript that addresses the points raised during the review process.

In their thorough and thoughtful critiques the Reviewers have raised several areas of required clarification and elaboration in the methodology and results.  This should be addressed in a MAJOR REVISION to the submitted manuscript.

We look forward to receiving your revised manuscript.

Kind regards,

Scott Brakenridge, M.D.

Academic Editor

PLOS ONE

Journal Requirements:

2)  Thank you for stating the following in the Acknowledgments Section of your manuscript:

[The MIMIC database is funded by the National Institute of Health through the NIBIB R01 grant EB017205.]

 [The authors received no specific funding for this work.]

Reviewers' comments:

Reviewer's Responses to Questions

**Comments to the Author**

1. Is the manuscript technically sound, and do the data support the conclusions?

Reviewer #1: Yes

Reviewer #2: Yes

2. Has the statistical analysis been performed appropriately and rigorously? 

Reviewer #1: No

Reviewer #2: No

3. Have the authors made all data underlying the findings in their manuscript fully available?

Reviewer #1: Yes

Reviewer #2: Yes

4. Is the manuscript presented in an intelligible fashion and written in standard English?

Reviewer #1: Yes

Reviewer #2: Yes

5. Review Comments to the Author

Reviewer #1: The authors present a retrospective analysis comparing accuracy of ProVent14 vs. a neural network in predicting 3-month mortality among ICU (MIMIC III) patients who require 7 or more days of mechanical ventilation. The authors frame this approach on the concept that most tracheostomies are performed after 7-10 days of mechanical ventilation, but 60-70% of such patients will die within one year and only 10% achieve functional independence. I agree that under these circumstances, accurate prognostic information may be particularly useful to patients, caregivers, and clinicians involved in the decisions regarding early tracheostomy. The research question is relevant but the methods for addressing it are suboptimal. The neural network outperforms ProVent14, but ProVent performed quite poorly (AUC 0.59 vs. 0.74 for the neural network). It is difficult to ascertain why ProVent performed so poorly in this analysis, given that it's discrimination was much higher on prior external validation (AUC 0.75). Was the implementation of the ProVent14 model in the present study identical to implementation in the description by Udeh (Ann Am Thorac Soc 2015)? If so, can the authors speculate as to why there was such significant performance degredation between the previously published external validation and their application of ProVent? Even if the implementation was the same, I would recommend that the authors compare the neural network to a stronger model, even if it is not designed specifically for mechanically ventilated patients. Logistic regression models have achieved greater discrimination (Hadique Ann Am Thorac Soc 2017, AUC 0.84) in predicting long-term mortality among ICU patients.

Reviewer #2: The authors present a neural network based model for predicting long-term mortality in patients on mechanical ventilation. The manuscript has been written well, and the results presented clearly. The authors provide a strong motivation to justify the need for a mortality prediction model for patients on mechanical ventilation.

My concerns regarding the manuscript are as follows:

1) More details regarding the neural network architecture is required. How many layers? What was the learning rate? What was the dropout factor? What was the L1-L2 parameter? How were each of the above hyper-parameters chosen?

2) How does the calibration curve look for the proposed model on the training and testing cohorts?

3) Do the authors believe their proposed model is generalizable to other institutions? Do the others believe that constructing a model based on 250 clinical variables, could easily be transported to other medical institutions?

4) What was the performance of the ProVent model constructed using a neural network instead of logistic regression?

5) How was normalization of data performed?

6) Could the authors compare their proposed model against the model developed by Sun et. al [1]?

7) Could the authors provide area under the precision recall curve as well in addition to the AUC plots? What was the PPV and NPV of their proposed model?

[1] Sun, Y., Li, S., Wang, S., Li, C., Li, G., Xu, J., Wang, H., Liu, F., Yao, G., Chang, Z. and Liu, Y., 2020. Predictors of 1-year mortality in patients on prolonged mechanical ventilation after surgery in intensive care unit: a multicenter, retrospective cohort study. BMC anesthesiology, 20(1), pp.1-9.

6. PLOS authors have the option to publish the peer review history of their article (what does this mean?). If published, this will include your full peer review and any attached files.

Reviewer #1: No

Reviewer #2: No

---

## [Author Response · Author response to Decision Letter 0]

28 Jan 2021

Response to Reviewers

Reviewer #1: 

1. The authors present a retrospective analysis comparing accuracy of ProVent14 vs. a neural network in predicting 3-month mortality among ICU (MIMIC III) patients who require 7 or more days of mechanical ventilation. The authors frame this approach on the concept that most tracheostomies are performed after 7-10 days of mechanical ventilation, but 60-70% of such patients will die within one year and only 10% achieve functional independence. I agree that under these circumstances, accurate prognostic information may be particularly useful to patients, caregivers, and clinicians involved in the decisions regarding early tracheostomy. The research question is relevant but the methods for addressing it are suboptimal. The neural network outperforms ProVent14, but ProVent performed quite poorly (AUC 0.59 vs. 0.74 for the neural network). It is difficult to ascertain why ProVent performed so poorly in this analysis, given that its discrimination was much higher on prior external validation (AUC 0.75). 

Response: Thank you for your observations. As the reviewer notes, the ProVent Score has been validated for prediction of mortality based on data collected on both day 14 and day 21. In our investigation the ProVent score generated a lower AUC than was found in previous investigations. The discrepancy between our results and previous investigations is due to the fact that our model was built to predict mortality based on data and clinical features collected at day 7, rather than day 14 or day 21. The ProVent model is not well suited for predictions based on day 7 data. The clinical features of most disease processes evolve over time; therefore, it is not uncommon for the test characteristics of a clinical prediction model to degrade when the model is applied at a different point in time than it had been originally developed. There are many instances of similar phenomenon in clinical medicine and research. For instance, a patient’s neurological exam immediately after cardiac arrest is not a reliable predictor of poor neurological outcome, but a neurological exam 72 hours after cardiac arrest becomes much more reliable. (Sandroni 2018) Thus prediction models developed for 72 hours post cardiac arrest do not perform well when applied immediately after cardiac arrest. In the case of ICU mortality prediction modeling, multiple ICU mortality prediction scores, such as the APACHE, SAPS II, and MODS scores, (all of which were developed based on the first 24 hours of ICU admission), have been tested and found to be less accurate predictors of mortality when applied at earlier time points, such as arrival to the emergency department. (Nguyen, 2000) The discussion section has been amended to more clearly explain why the ProVent score performed poorly in this cohort. 

1. Sandroni, C., D’Arrigo, S. & Nolan, J.P. Prognostication after cardiac arrest. Crit Care 22, 150 (2018). https://doi.org/10.1186/s13054-018-2060-7

2. Nguyen HB, Rivers EP, Havstad S, Knoblich B, Ressler JA, Muzzin AM, Tomlanovich MC: Critical care in the emergency department: a physiologic assessment and outcome evaluation. Acad Emerg Med 2000, 7: 1354-1361.

2. Was the implementation of the ProVent14 model in the present study identical to implementation in the description by Udeh (Ann Am Thorac Soc 2015)? If so, can the authors speculate as to why there was such significant performance degradation between the previously published external validation and their application of ProVent? 

Response: In designing this study we wanted to give the ProVent Model the ‘best’ chance to perform well at day 7. Besides the time-point of data collection, the other fundamental difference between our model and the ProVent model is that our model is based on deep learning and ProVent is based on a multivariate logistic regression. We first attempted to validate the ProVent Score using the same point system described in the original ProVent article, but using data collected at day 7 rather than day 14 or 21. (This included first developing a cohort of patients receiving mechanical ventilation for 7 days or longer, then ascertaining the presence or absence of the four ProVent Variables on day 7). As with the original ProVent studies, need for dialysis was deemed present if the patients received dialysis on or within 48 hours of day 7, and age was categorized into three groups (18–49, 50–64, and 65+ years). This modeling yielded poor results for the ProVent Score. In effort to give the ProVent model the ‘best’ chance possible, we next optimized the ProVent logistic regression model to include all of the variables available to the deep learning model. Rather than limiting the logistic regression ProVent Model to only include the 4 variables that it was originally developed with, we also ran a logistic regression model using all of the variables we had collected and made available to the deep learning model. While this did improve the performance of the ProVent model, it was not enough to outperform the deep learning model, as we have reported. The methods section of the manuscript has been amended to address the reviewer’s concern. 

3. Even if the implementation was the same, I would recommend that the authors compare the neural network to a stronger model, even if it is not designed specifically for mechanically ventilated patients. Logistic regression models have achieved greater discrimination (Hadique Ann Am Thorac Soc 2017, AUC 0.84) in predicting long-term mortality among ICU patients.

Response: Thank you for highlighting this important study by Hadique et al. Their investigation, along with several other generalized ICU mortality prediction models (most notably the APACHE score), boasts a high AUC. However, the poor accuracy of general ICU mortality prediction scores for patients facing prolonged mechanical ventilation was one of the central motivations to develop the ProVent Score, and is our motivation as well.(Carson 2012) 

In this investigation we did not compare our model head-to-head with any of the many general ICU mortality prediction scores for two reasons. First, most general ICU mortality prediction scores use in-hospital mortality as their endpoint (such as the APACHE score). General ICU mortality prediction scores do not perform well in predicting long term survivorship, and thus leave considerable uncertainty around long term prognosis. (Angus, 2000) This knowledge gap was identified by the ProVent investigators and was the central motivation for developing the ProVent score. (Carson ) We agree with the ProVent investigators that the lack of accurate prognostic estimates for long term survival presents an important clinical challenge, and thus we undertook the present investigation. 

Secondly, nearly all well-validated ICU mortality prediction models are developed based on the first 24 hours of ICU patient data (this is true of the Hadique model that the reviewer highlights). These scores tend to perform poorly when applied later on during the ICU stay because the patient cohort shifts considerably; many ICU survivors are discharged from the ICU prior to 7 days, and many ICU descendants have died prior to 7 days. Additionally, the natural history of disease evolves during the ICU course and variables and their impact on mortality shift. Of those patients with similarly bad APACHE scores on day one of their ICU stay, some will survive and some will die. On day 7, 14, or 21 of the ICU stay those that will survive and those that will die have become further differentiated, requiring a re-calibrated score. 

In terms of the investigation highlighted by Hadique, their score has three elements, each obtained on day one of ICU; (1) the APACHE Score, (2) the Charlson Comorbidity Score (CCS), and (3) the providers impression of 6-month mortality obtained on day 1 of ICU stay (the ‘surprise question’) The variables from the first two elements, day 1 APACHE and CCS, are almost completely incorporated into our model (the difference being our model used the Elixhauser comorbidity score rather than CCS because of the Elixhauser score’s demonstrated superiority in mortality prediction). These variables were included in a logistic regression model (see response above) which underperformed compared to our deep learning model. The third element of the Hadique model, the ‘surprise question’ must be obtained prospectively, thus we are unable to generate a direct comparison in this investigation. 

1. Carson SS, Bach PB. Predicting mortality in patients suffering from prolonged critical illness: an assessment of four severity-of-illness measures. Chest. 2001 Sep; 120(3):928-33.

2. Angus DC, Clermont G, Kramer DJ, Linde-Zwirble WT, Pinsky MR. Short-term and long-term outcome prediction with the Acute Physiology and Chronic Health Evaluation II system after orthotopic liver transplantation. Crit Care Med [Internet]. 2000 Jan;28(1):150–6

Reviewer #2: 

The authors present a neural network-based model for predicting long-term mortality in patients on mechanical ventilation. The manuscript has been written well, and the results presented clearly. The authors provide a strong motivation to justify the need for a mortality prediction model for patients on mechanical ventilation.

My concerns regarding the manuscript are as follows:

1) More details regarding the neural network architecture is required. How many layers? What was the learning rate? What was the dropout factor? What was the L1-L2 parameter? How were each of the above hyper-parameters chosen?

Response: The neural network consisted of three layers: an input layer, a single hidden layer consisting of 100 neurons, and an output layer. A constant learning rate of 0.001 was used with the Adam optimizer. There was no dropout. L2 regularization was performed with a penalty value of 0.0001.

A sigmoid activation function was employed in the output layer, with which the output range was interpreted as a prediction probability of the modeled outcome. Neural architecture searches and hyperparameter optimization were not performed. Instead, common default parameters were chosen purposefully, as they yielded acceptable performance.

2) How does the calibration curve look for the proposed model on the training and testing cohorts?

Response: Thank you for addressing this reporting gap. The testing and training calibration curves have been added as S4 Fig. 

3) Do the authors believe their proposed model is generalizable to other institutions? Do the others believe that constructing a model based on 250 clinical variables, could easily be transported to other medical institutions?

Response: Thank you for addressing this important topic. In developing our model, we purposefully selected variables that would be readily available across any modern ICU and routinely collected during processes of care. Therefore, we expect the input values for the model would not represent a barrier to implementation. Like most machine learning predictive models, the complexity of our model necessitates automated implementation in the EMR to be useful - meaning that adopters of the model would install the algorithm into the EMR to automatically generate a prediction. We plan to embark upon implementation once external validation is complete. 

4) What was the performance of the ProVent model constructed using a neural network instead of logistic regression?

Response: The ProVent model was constructed using the 4 variables from the initial citation using logistic regression (see previous response). In addition, in order to give the ProVent Model the ‘best shot’ at outperforming our deep learning model, we attempted to optimize the ProVent model by running a multivariate regression with all of the variables available to the deep learning model. Both the prescribed ProVent model and the modified logistic regression failed to outperform the machine learning model. The manuscript methods section has been amended to make the clearer. 

5) How was normalization of data performed?

Response: Continuous variables were standardized to a mean of zero and scaled to unit variance. This has been added to the methods section to improve clarity.

6) Could the authors compare their proposed model against the model developed by Sun et. al [1]?

Response: Thank you for highlighting this interesting study. For their investigation, Sun et. al. examined factors predicting 1-year mortality among a cohort of 124 Chinese post-operative surgical patients requiring mechanical ventilation for greater than 21 days using a logistic regression model. Their model ultimately incorporated most of the same predictors from the landmark work by Carson, the ProVent Score. Specifically, Sun found that cancer diagnosis, no tracheostomy, enteral nutrition intolerance, blood platelet count ≤150 × 109/L, vasopressor requirement, and renal replacement therapy on the 21st day of MV were associated with shortened 1-year survival. Thus, with the exception of cancer diagnosis and enteral nutrition intolerance, these are the same factors looked at by Carson in the ProVent score. Similar to the ProVent score, this model is limited in that its application comes at 21 days, rather than 7 days, thus leading to a similar dearth of prognostic information earlier in the patient’s course, when clinicians and families are facing decisions around care trajectory (i.e. to place or not place a tracheostomy). This study, though valuable, does not unseat the ProVent score as the dominant model. First, the study is small, and limited to a cohort of Chinese patients. Second, whereas the ProVent Score has been validated numerous times among both general ICU and surgical ICU patients, this score by Sun et al has not yet been externally validated. We recognize that there is the potential for the model by Sun to be externally validated and potential for it to one day outperform the ProVent Score. Reassuringly, our model includes the need for parenteral nutrition and underlying cancer diagnosis. Future research will compare our model to other predictors such as this. 

7) Could the authors provide area under the precision recall curve as well in addition to the AUC plots? What was the PPV and NPV of their proposed model?

Response: Thank you for highlighting this reporting gap. The positive and negative predictive value for the neural network at 3 months was 0.64 and 0.72 respectively, and 0.67 and 0.71 at 12 months. This has been added to the results section. We have added the area under the precision recall curve as supplementary figure 5 (S5 Fig).

---

## [Decision Letter · Decision Letter 1]

22 Mar 2021

PONE-D-20-32346R1

Deep Learning to Predict Long-term Mortality in Patients Requiring 7 days of Mechanical Ventilation

PLOS ONE

Dear Dr. George,

Thank you for submitting your manuscript to PLOS ONE. After careful consideration, we feel that it has merit but does not fully meet PLOS ONE’s publication criteria as it currently stands. Therefore, we invite you to submit a revised version of the manuscript that addresses the points raised during the review process.

**There are a few minor points recommended by the reviewers to be addressed.  A full reply to reviewer letter is not required, just the referred to manuscript revisions.**

We look forward to receiving your revised manuscript.

Kind regards,

Scott Brakenridge, M.D.

Academic Editor

PLOS ONE

Journal Requirements:

Reviewers' comments:

Reviewer's Responses to Questions

**Comments to the Author**

1. If the authors have adequately addressed your comments raised in a previous round of review and you feel that this manuscript is now acceptable for publication, you may indicate that here to bypass the “Comments to the Author” section, enter your conflict of interest statement in the “Confidential to Editor” section, and submit your "Accept" recommendation.

Reviewer #1: All comments have been addressed

Reviewer #2: (No Response)

2. Is the manuscript technically sound, and do the data support the conclusions?

Reviewer #1: (No Response)

Reviewer #2: Yes

3. Has the statistical analysis been performed appropriately and rigorously? 

Reviewer #1: (No Response)

Reviewer #2: Yes

4. Have the authors made all data underlying the findings in their manuscript fully available?

Reviewer #1: (No Response)

Reviewer #2: Yes

5. Is the manuscript presented in an intelligible fashion and written in standard English?

Reviewer #1: (No Response)

Reviewer #2: Yes

6. Review Comments to the Author

Reviewer #1: (No Response)

Reviewer #2: The authors have resolved most of my concerns. I have one final comment which if included might strengthen the paper:

1) The authors mention that they trained a multivariate regression model with same number of variables available to the deep learning model. I do not see the performance of this model reported anywhere in the manuscript. It might be helpful to the reader if the results of a logistic regression model constructed using the same variables as the deep learning model is reported.

7. PLOS authors have the option to publish the peer review history of their article (what does this mean?). If published, this will include your full peer review and any attached files.

Reviewer #1: **Yes: **Tyler Loftus

Reviewer #2: No

---

## [Author Response · Author response to Decision Letter 1]

11 Apr 2021

Reviewer Comment and Author Response:

Reviewer #2: "The authors have resolved most of my concerns. I have one final comment which if included might strengthen the paper: The authors mention that they trained a multivariate regression model with the same number of variables available to the deep learning model. I do not see the performance of this model reported anywhere in the manuscript. It might be helpful to the reader if the results of a logistic regression model constructed using the same variables as the deep learning model is reported."

Response: Thank you for your comment. The reviewer correctly notes that the reporting in one portion of the methods section should be clearer. To remedy this, and to ensure that the results are consistent with the reported methods, we ran two additional models using logistic regression techniques that utilization all 250 variables available to the neural network. The methods section and results section of the manuscript have been further clarified in the revised draft, and the discussion has been updated to reflect these latest findings. Results for the additional logistic regression models are included as supplemental figure 6 (S6 FIG).

---

## [Decision Letter · Decision Letter 2]

7 Jun 2021

Deep Learning to Predict Long-term Mortality in Patients Requiring 7 Days of Mechanical Ventilation

PONE-D-20-32346R2

Dear Dr. George,

We’re pleased to inform you that your manuscript has been judged scientifically suitable for publication and will be formally accepted for publication once it meets all outstanding technical requirements.

Kind regards,

Yu Ru Kou, PhD

Academic Editor

PLOS ONE

Additional Editor Comments (optional):

Reviewers' comments:

Reviewer's Responses to Questions

**Comments to the Author**

1. If the authors have adequately addressed your comments raised in a previous round of review and you feel that this manuscript is now acceptable for publication, you may indicate that here to bypass the “Comments to the Author” section, enter your conflict of interest statement in the “Confidential to Editor” section, and submit your "Accept" recommendation.

Reviewer #1: All comments have been addressed

2. Is the manuscript technically sound, and do the data support the conclusions?

Reviewer #1: Yes

3. Has the statistical analysis been performed appropriately and rigorously? 

Reviewer #1: Yes

4. Have the authors made all data underlying the findings in their manuscript fully available?

Reviewer #1: Yes

5. Is the manuscript presented in an intelligible fashion and written in standard English?

Reviewer #1: Yes

6. Review Comments to the Author

Reviewer #1: (No Response)

7. PLOS authors have the option to publish the peer review history of their article (what does this mean?). If published, this will include your full peer review and any attached files.

Reviewer #1: **Yes: **Tyler Loftus

---

## [Editor Report · Acceptance letter]

10 Jun 2021

PONE-D-20-32346R2 

Deep learning to predict long-term mortality in patients requiring 7 days of mechanical ventilation 

Dear Dr. George:

I'm pleased to inform you that your manuscript has been deemed suitable for publication in PLOS ONE. Congratulations! Your manuscript is now with our production department. 

Kind regards, 

on behalf of

Dr. Yu Ru Kou 

Academic Editor

PLOS ONE